# Wearable High Voltage Compliant Current Stimulator for Restoring Sensory Feedback

**DOI:** 10.3390/mi14040782

**Published:** 2023-03-30

**Authors:** Riccardo Collu, Roberto Paolini, Martina Bilotta, Andrea Demofonti, Francesca Cordella, Loredana Zollo, Massimo Barbaro

**Affiliations:** 1Department of Electrical and Electronics Engineering, University of Cagliari, Piazza D’Armi, 09123 Cagliari, Italy; 2Research Unit of Advanced Robotics and Human-Centred Technologies (CREO Lab), Università Campus Bio-Medico di Roma, 00128 Rome, Italy

**Keywords:** neurostimulator, sensory feedback, wearable electronic system, TENS

## Abstract

Transcutaneous Electrical Nerve Stimulation (TENS) is a promising technique for eliciting referred tactile sensations in patients with limb amputation. Although several studies show the validity of this technique, its application in daily life and away from laboratories is limited by the need for more portable instrumentation that guarantees the necessary voltage and current requirements for proper sensory stimulation. This study proposes a low-cost, wearable high-voltage compliant current stimulator with four independent channels based on Components-Off-The-Shelf (COTS). This microcontroller-based system implements a voltage-current converter controllable through a digital-to-analog converter that delivers up to 25 mA to load up to 3.6 kΩ. The high-voltage compliance enables the system to adapt to variations in electrode-skin impedance, allowing it to stimulate loads over 10 kΩ with currents of 5 mA. The system was realized on a four-layer PCB (115.9 mm × 61 mm, 52 g). The functionality of the device was tested on resistive loads and on an equivalent skin-like RC circuit. Moreover, the possibility of implementing an amplitude modulation was demonstrated.

## 1. Introduction

Amputation of a limb due to traumatic events or the course of disease negatively affects an individual’s quality of life. It is estimated that more than 215,000 amputation surgeries are performed each year [1]. This number is estimated to increase due to the aging of the population and the corresponding increased incidence of diabetes and vascular disease [2,3].

Notwithstanding the huge advancement in the development of advanced polyarticulated prosthetic hands aiming to replicate the functions of human hands [4,5] and prosthetic leg seeking to provide a natural support, propulsion, flexibility, and load relief [6], the current limb prostheses do not provide sensory feedback to the user [7,8]. Relying exclusively on sight and tactile information from stump-socket interface, people with amputations are not able to execute confident grip forces and undertake fine manipulations [8] or have to face off with deambulation disorders [9]. This leads to an abandoning rate of upper and lower limb prosthesis of 39% and 60%, respectively [8,10].

Several solutions have been proposed in literature to restore sensory feedback in amputees [11,12,13,14,15,16,17,18,19,20,21].

The use of invasive interface with Peripheral Nervous System have shown promising results thanks to their selectivity in the elicitation of somatotopic tactile sensations, but they present disadvantages related to invasiveness due to surgery, fibrotic reaction, and weak long-term stability [22].

Transcutaneous Electrical Nerve Stimulation (TENS) is a non-invasive valid alternative since it uses external electrodes placed on the skin to stimulate the underlying nerves evoking referred tactile sensations. Although TENS is well established for its analgesic effect [23], it represents an emerging technology in the field of upper and lower limb prosthetics [24,25]. 

Osborn et al., developed a bidirectional prosthesis based on TENS and an innovative electronic dermis (e-dermis), allowing to elicit tactile information on the missing hands of a 29-year-old bilateral amputated [26]. D’Anna et al., investigated the effect of TENS on induced sensations in four amputees. They observed that even if most of the sensations were of paresthesia, the stimulation improved subjects’ control of a prosthetic hand [27]. Pan et al., used a high-density electrode grid to induce haptic sensation on the lower limb of five trans-radial amputee subjects, showing the possibility of eliciting sensation in different regions of the foot [28]. Wang et al., investigated the effectiveness of TENS in sensory feedback by observing the activity induced on the electroencephalogram [29]. Scarpelli et al., developed three encoding algorithms for the elicitation of slippage sensations in the hand of nine healthy subjects [24].

Most of the previous studies were performed using commercial electrical stimulators. Although such devices are versatile and present a high level of programmability, they are cumbersome and expensive. Subsequently, in recent years, different portable electrical stimulators have been designed and developed in the literature. 

Electrical stimulation can be executed in Voltage Mode Stimulation (VMS) or Current Mode Stimulation (CMS). In VMS, the stimuli are delivered by applying a known voltage across the electrodes. In CMS, the stimuli are delivered by applying a known current through the electrodes, i.e., in the load. CMS is typically preferred to VMS since the charge injected into the tissues does not depend on the load’s impedance, as shown in Figure 1. Nerve stimulation is typically performed through charge-balanced biphasic waveforms to avoid a dangerous accumulation of charge that can lead to cause damage to tissues [30].

Shirafkan et al., proposed a current stimulator based on a Flyback converter with an H-bridge with a voltage compliance of 80 V [31]. Masdar et al., developed a microcontroller-based current stimulator using a voltage-to-current converter to deliver the current to the load [32]. Cheng et al., proposed a circuit based on a step-up transformer directly connected to the load [33].

A wearable stimulator should be designed to have high programmability in terms of amplitude, frequency, and duration of stimulation. The simulator should deliver up to 10 mA, adapting to possible variations, and impedance ranges of human skin [34,35,36]. 

In this regard, a wearable electronic stimulator for TENS based on Components-Off-The-Shelf (COTS) is proposed. The developed system has been designed on a four-layer PCB with a voltage compliance of ±90 V, allowing it to stimulate currents up to 25 mA with a load of 3.6 kΩ. 

## 2. Materials and Methods 

THE SYSTEM: The system implements a Microcontroller Unit (MCU) through Espressif ESP32-DEVKIT32E, a development platform based on ESP32-WROOM-32UE dual-core 32-bit microprocessor with a built-in Bluetooth module and USB peripheral. The ESP32-DEVKIT32E Bluetooth module has a receiver with a declared sensitivity of −89 dBm and transmitter with selectable power between −12 dBm and +9 dBm, supporting class-1, class-2, and class-3 transmit output power. The design comprises four main units: (i) an MCU; (ii) Digital-to-Analog Converter (DAC); (iii) current stimulating output channels; and iv) voltage regulation, as shown in Figure 2. The four channels DAC was realized using Maxim Integrated (San Jose, CA, USA) MAX537, a quad channel, SPI-controlled, 12-bit voltage buffered DAC based on R-2R architecture. The use of Texas Instruments (Dallas, TX, USA) high-voltage amplifiers OPA462IDDA within the output channels guarantees an output current of up to 45 mA, allowing a good margin over the maximum 25 mA for which the system is designed. The device has two power domains: the low voltage domain is managed using the 5 V provided by the external power bank, the −5 V provided using Maxim Integrated switched-capacitor voltage inverter MAX870EUK + T, and the 2.5 V provided by low-dropout (LDO) voltage regulator Microchip Technology (Chandler, AZ, USA) TC2117-2.5VDBTR. The high voltage domain, which provides the ±90 V supplied in the output stage, was implemented using two different circuits based on the switching voltage regulators Analog Devices Inc. (Wilmington, MA, USA) LT8365. LT8365 is a voltage converter that obtains both positive and negative output voltages, guaranteeing a low quiescent current and low ripple on the output.

OUTPUT CHANNELS: A voltage-programmable voltage-to-current converter delivers the current to the load employing two amplifiers for each channel, low voltage rail-to-rail amplifier STMicroelectronics (Geneve, CH) TS462CST and high voltage Texas Instruments OPA462IDDA, as reported in Figure 3. TS462CST implements a differential amplifier that subtracts the value of the DAC’s output from the reference voltage to generate bipolar voltages in the range of ±2.5 V. According to a known resistor *R_stim_*, the differential amplifier’s output voltage sets the current delivered to the load. The subject that should be stimulated is connected to the feedback of the OPA462, and the direction of the current (e.g., the polarity of stimulation) depends on the output of the first stage. The value of the stimulating current $I_{stim}$ is fixed by the relation:(1)$$I_{stim}=\frac{V_{1}}{R_{stim}}$$

FIRMWARE AND MATLAB GUI: The firmware was designed to allow the device to communicate using USB and Bluetooth. At startup, the MCU initializes and configures the SPI registers, establishing communication with the DAC. The DAC’s output channels are initialized to provide no current in the output of the stimulation channels. Then, the device goes into a waiting state in which it waits for a command. When a command is received, the device starts the stimulation generating biphasic asymmetrical charge-balanced stimuli according to the pulse amplitude, pulse width, interphase delay, frequency, and stimulation time indicated in the command. 

Charge balancing is guaranteed by a control implemented inside the firmware. When a biphasic symmetrical waveform is required, the firmware computes the same duration for the anodic and cathodic phases, applying the same amplitude to both the phases:(2)$$Q=Q_{cathodic}=PW_{cathodic}*A_{cathodic}=PW_{anodic}*A_{anodic}=Q_{anodic}$$

When an asymmetric stimulation waveform is required, the firmware sets a predefined ratio between the anodic and cathodic phase to 1/10. In case of asymmetric waveform with higher cathodic amplitude than anodic amplitude, the firmware sets the anodic amplitude to $A_{anodic}=$ 0.1 × $A_{cathodic}$. To guarantee the same charge in the two stimulation phases, the firmware computes the duration of the anodic phase as [37]:(3)$$PW_{anodic}=10*\frac{Q}{A_{anodic}}$$

A custom MATLAB Graphical User Interface (GUI) was designed and developed in MATLAB 2021b. The interface allows establishing a Bluetooth or USB connection with the device and allows the possibility to select the stimulation parameters and the channels to activate.

Dual control on the parameters is performed in the MATLAB interface and MCU firmware, avoiding the delivery of current stimuli with duration or amplitude outside the desired ranges [38].

## 3. Results and Discussion

The developed system was a 115.9 mm × 61 mm four-layer PCB: see Figure 4. The total mass of the device is about 52 g. The power consumption of the device has been measured and results of 700 mW.

We tested the device by powering it with a phone power bank with capacitance of 36,000 mAh and using resistive loads. Given the consumption of the device, it would be possible to use a 2000 mAh power bank and still guarantee a 12 h run time. Measurements were taken using the oscilloscope MSO654A produced by Agilent Technologies (Santa Clara, CA, USA).

The capability to generate different waveforms using a 10 kΩ was investigated as reported in Figure 5. Then, the system was tested also with loads of 1 kΩ and 4.7 kΩ, as reported in Appendix A.

To test the device in a more realistic condition, we connected the output electrodes to a “Skin-Like Circuit” composed by a capacitor *C_es_* in parallel to a resistor R_es_ and a series resistor *R_s_*. The values of the components were chosen according to [35,39]: therefore, *C_es_* = 47 nF, *R_es_* = 22 kΩ, and *R_s_* = 470Ω. The device stimulation capabilities are reported in Figure 6.

We also investigated the possibility of generating stimulation patterns with amplitude modulation like in the encoding algorithm tested in [40,41], as reported in Figure 6C. To compute the current delivered by the stimulator, we measured the voltage across the resistor *R_es_* and divided it by the value of the resistor during data elaboration on MATLAB.

As highlighted by the tests, the high voltage compliance allows the system to adapt to different impedance conditions of the electrode-skin contact, modeled, initially, by a resistive load, then using a skin-like circuit. The possibility of programming the stimulation waves via SPI communication and the voltage control offered by the stimulation channel make the device easily applicable to deliver different stimulation patterns. Furthermore, the possibility of implementing amplitude modulation makes the system compatible for the restoration of tactile sensations in patients with limb amputation [33,39].

In an upper limb stimulation, the four available channels would allow the three nerves that graft the hand (ulnar, radial, and median) to be stimulated simultaneously, thus offering a tool for the investigation of algorithms that allow complete restoration of the hand’s sensations, also allowing to find the best electrode combinations according to the subject’s needs. Moreover, the high programmability and resolution of the device would enable investigating of more sophisticated algorithms capable of mimicking the natural behavior of skin receptors, improving the quality of induced sensations in the missing limb, and leading to the advancement of the quality and usability of prosthetic systems as suggested in [42,43,44].

In Table 1**.**, we reported a comparison of our device with selected stimulators reported in the literature.

The proposed architecture is extremely simple and requires few components for each stimulation channel, only two amplifiers, easily reducible using a bipolar DAC, which would avoid implementing the subtraction circuit we used to generate the bipolar waves, reducing device encumbrance. The size of the device is, in fact, another critical point that should be improved. Although the dimensions of our device are not notably different from those reported in several literature examples [31,46], the full integration into a prosthetic leg or arm system requires further scaling of the size. To do this, the complete integration of the microcontroller inside the PCB and replacing the terminal blocks used for the electrodes in favor of using jack cables would drastically reduce the device encumbrance.

Although our approach exhibits power consumption consistent with what is found in the literature [46,48] and a substantially higher voltage compliance, a further improvement could be achieved through optimization of the voltage regulation circuits.

In this regard, although using off-the-shelf components enables the realization of a relatively low-cost device, it is the main bottleneck in optimizing power consumption. To maximize the performance of a sensory feedback device, a custom chip with high-voltage compliance should be developed.

## 4. Conclusions

In this work, we proposed a wearable four-channel high-voltage compliance (±90 V) current stimulator based on components of the shelf. The system has been realized using four main units: a microcontroller unit (MCU), Digital-to-Analog Converter (DAC), current-stimulation channels, and voltage regulation. Two high-voltage regulators, Analog Devices LT8365, were used to provide the ±90 V required by the current-stimulation channels. The Maxim Integrated MAX870EUK + T and Microchip TC2117 provided 5 V and 2.5 V used by the rest of the device. The system features high programmability through a programmable stimulation channel controlled by the four-channel DAC, Maxim Integrated MAX537, programmable from the MCU via SPI protocol. The architecture of current stimulation channels makes it easy to generate different stimulation patterns, e.g., with amplitude modulation, currently implemented in prosthetic applications. The design process of the device was done to achieve working ranges compliant with the sensory feedback restoring requirements reported in the literature. The device shows a power consumption of 700 mW, that needs to be improved, and has a weight of 52 g. The device allows to implement several kinds of waveforms, and the capability to communicate with USB or Bluetooth with an external device allows to integrate and test the system with a prosthetic system equipped with pressure sensors.

Further studies will be devoted to validating the developed system by extensively comparing its performance with the one obtained with a gold-standard commercial electrical stimulator. Moreover, the system will be miniaturized and tested on healthy subjects.

## Figures and Tables

**Figure 1 micromachines-14-00782-f001:**
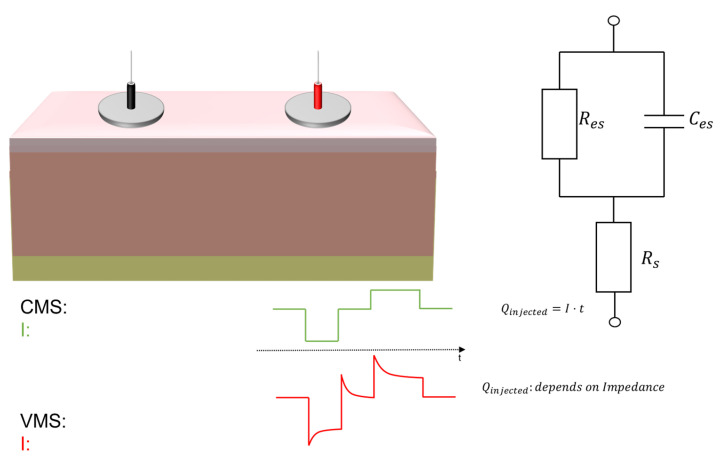
Equivalent skin-electrodes model. The contact between the electrodes and the skin can be modeled using a parallel circuit composed of a resistor *R_e_*_s_ and a capacitor *C_es_* in series with a resistor *R_s_*. In VMS, the current profile is affected by the charge and discharge of the capacitor *C_es_*, leading to an uncertainty on the total delivered charge. In CMS, the charge delivered depends only on the shape of the current profile, e.g., using a rectangular profile, the delivered charge is given by the product between the current and the duration of the stimulation.

**Figure 2 micromachines-14-00782-f002:**
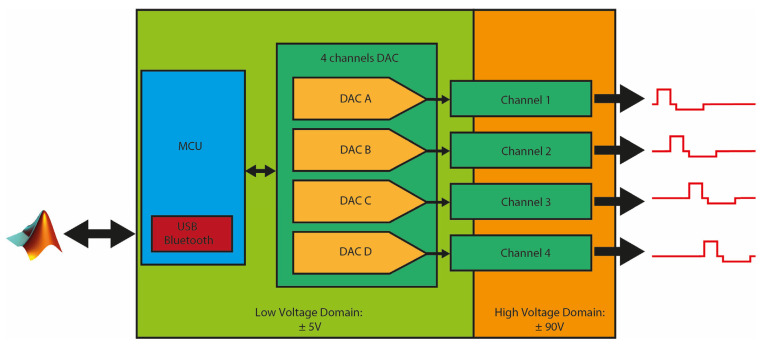
Block diagram of the proposed system. The platform communicates with MATLAB using MCU built-in USB or Bluetooth peripherals. The MCU controls the four channels DAC allowing to set the desired current in output in each output channel. MCU and DAC work in the low-voltage domain (±5 V), while the output channels operate in the high-voltage domain (±90 V).

**Figure 3 micromachines-14-00782-f003:**
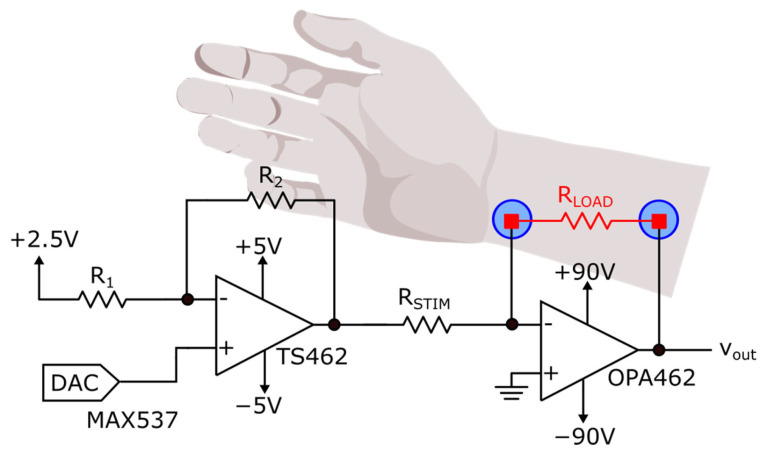
Stimulation channel. The stimulation circuit is developed in two stages: a low-voltage one implemented using TS462CST, and a high-voltage one implemented using OPA462IDDA. The low-voltage stage adopts a subtractor circuit that removes the reference voltage (2.5 V) from the output voltage of the DAC, thus allowing biphasic waveforms to be generated. The output of the subtractor fixes the current that will be delivered in the second stage based on the value of a known resistor R. The high voltage compliance of the second stage allows it to deliver up to 25 mA with loads up to 3.6 kΩ.

**Figure 4 micromachines-14-00782-f004:**
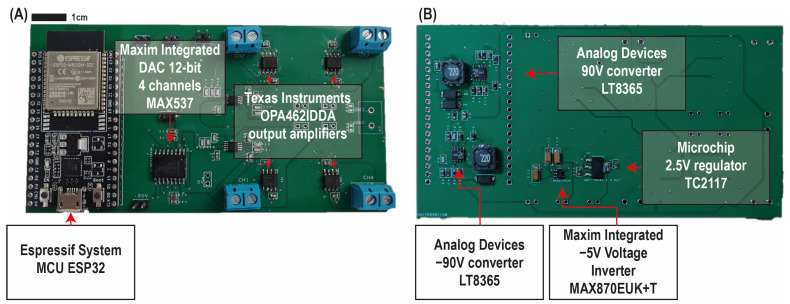
The developed PCB: frontal (**A**) and rear (**B**) view. High-voltage and low-voltage regulators have been located in the bottom layer of the PCB. The MCU, DAC, and amplifiers have been located in the top layer of the PCB.

**Figure 5 micromachines-14-00782-f005:**
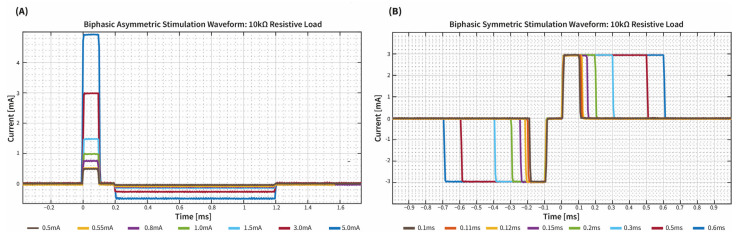
Waveform generation capability on 10 kΩ resistive load using (**A**) Biphasic asymmetric waveform with different amplitude and pulse width 100 μs; (**B**) Biphasic symmetric waveform with different pulse width and amplitude 3 mA.

**Figure 6 micromachines-14-00782-f006:**
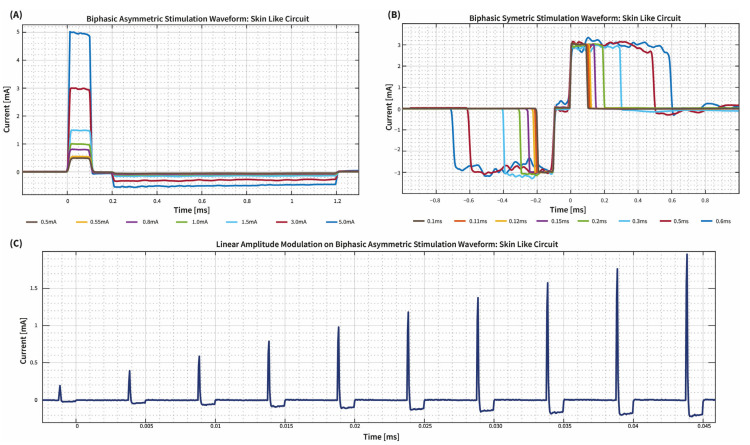
Stimulation on Skin-like circuit using (**A**) Biphasic asymmetric waveform with different amplitude and pulse width 100 μs (**B**) Biphasic symmetric waveform with different pulse width and amplitude 3.5 mA (**C**) Linear amplitude modulation with Biphasic asymmetric waveform in the range 0.2 mA–2 mA, pulse width of 100 μs, and frequency 200 Hz.

**Table 1 micromachines-14-00782-t001:** Comparison of our work with selected stimulators in the literature.

Work	Input Voltage	Output Voltage	Size	PW	Freq	Shape	Tested on
This work	5 V	±90	115.9 mm × 61 mm	≥50 μs	≤500 Hz	Programmable	Skin-like circuit
[45]	-	±72	46 mm × 89 mm	-	≤1 kHz	Sinusoidal	60 kΩ load
[31]	3.7–4.2 V	±80	96 mm × 89 mm	100–1000 μs	1–200 Hz	Biphasic Symmetric	RC series circuit R ≤ 2 kΩ C ≤ 1 μF
[46]	12 V	±60	170 mm × 75 mm	100–600 μs	20–80 Hz	Biphasic Asymmetric	1 kΩ
[47]	3.3 V	23	90 mm × 50 mm	0–500 μs	1–60 Hz	Biphasic	15 kΩ

## Data Availability

Not applicable.

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
