# Peer review of "Wearable High Voltage Compliant Current Stimulator for Restoring Sensory Feedback"

_micromachines, 2023, doi:10.3390/mi14040782_

Round 1

Reviewer 1 Report

This work presents a PCB-based stimulation system with off-the-shelf components. Unfortunately, this work is mainly a design project, without any clear justification for circuit design considerations. The measurement results are purely waveform, without any detailed analysis of the linearity/residual charge/voltage compliance, etc. The safety measure is missing for such high voltage stimulation. The overall contribution and target application of this work are not clear.

Some other components:

1. It seems simple resistors are used as tissue-electrode interfaces, this effect is not the real situation.

2. Please give detail about charge balancing, either algorithm or circuits.

3. Please provide measured results for negative current output. Fig 6 should be better organized to avoid lines overlapping.

Author Response

Thanks to the remarks of the reviewers, we thoroughly revised manuscript entitled "Wearable High Voltage Compliant Current Stimulator For Sensory Feedback Restoring" and submit it for revision.

We think that the paper, in this new form, is more readable, clear and unambiguous and we really want to thank the reviewers that, with their remarks and objections, helped us to improve the presentation and the interpretation of experimental results.

For the reviewers’ convenience, we prepared a modified version of the manuscript with all the modifications highlighted in red. In the following, we report the reviewers’ comments and our answers (in italic) and reference the position in the manuscript in which we modified the original document accordingly to the remarks or report the actual portions of text added to the original manuscript (in red).

You will find our answers in the attached PDF.

Reviewer 2 Report

This study proposed a low-cost, high-voltage compliant current stimulator and an accompanying GUI software. Overall, the authors made an electrical stimulator that works properly and can output a certain waveform. However, the content of the article only rarely covers the research addressed by the topic. The performance characterization of the circuit system is also inadequate. It is not recommended to publish this article if the following issues cannot be added or resolved.

1. The content of the article does not match the title. In other words, the content of the article does not demonstrate the significance of the research that the title hopes to convey.

       (1) According to the text, the electrical stimulator proposed in this paper is a large-size circuit system. How do you explain the 'wearable' in the title? At least it should have a safe, non-toxic and reliable package. Such a high supply voltage (90V, Figure 2) must require a relatively large power supply wired to this electrical stimulator, then it should be difficult to achieve wearability.

(2) The article does not contain the sensory feedback restoring described in the title and lacks an explanation of the application potential and basic qualitative validation results.

2. The paper shows that the circuit model for skin electrode contact impedance is a resistive-capacitive network (Figure 1). However, in verifying the circuit performance, only resistive loads are used (Line 135). Why didn’t you follow the model and did the test with resistive-capacitance loads? Such test results do not prove the potential of the application.

3. The load for testing is only below 10kΩ (Line 21, 138), yet not many electrodes can stably achieve skin contact impedance below 10kΩ. Why not do in vivo or ex vivo testing? If there is concern about the system causing harm to the organism, it can be tested with fresh pork with skin or other skin simulants. Please add a simple experiment and show valid results that are sufficient to support the topic.

4. This electrical stimulation system incorporated the ability to be controlled wirelessly via Bluetooth (Line 127, 128). But there is no elaboration on the transmission distance as well as power consumption, which are very important for wireless systems. In addition, the basic parameters of the whole electrical stimulation system were not described. Please add the parameters of the basic performance of the circuit, such as power consumption, transmission distance, signal-to-noise ratio, etc.

5. The direct capture of the oscilloscope test plot in Figure 5 is not very appropriate. Can you export the oscilloscope data and redo the graph?

6. In Figure 5, the time scale is unreasonably chosen, and the specific shape of the output waveform is not clearly seen, making readers difficult to determine the true duty cycle, rectangular wave distortion, etc. Please revise.

7. In Figure 6, it can be seen that the output waveform has a lot of noise or fluctuation, reaching close to 0.5mA at a load of 1k. Is this considered acceptable with a mere 5mA stimulus? Is it due to the ripple of the power supply? Or is it due to something else? Is it possible to assume that the circuit design is not perfect and lacks proper filtering and noise reduction? Please give a reasonable explanation, or optimize the circuit design.

8. Some of the stimulation waveforms of the electrical stimulator are shown in Figure 6. There are various types of electrical stimulation waveforms, and the results displayed are limited. Please add a richer test waveform.

9. A comparison of performance with non-portable or other portable TENS devices is missing in this paper. 

Author Response

(The authors gave the same response as above.)

Reviewer 3 Report

The idea very good to give feedback of sensory information to amputee's nervous system.

I have some concerns about the presented results:

- The implemented system was not compared to the other TENS systems. 

- It is not clear seeing the Conclusions what is the main novelty of this contribution. There are a few wearable TENS systems you can buy, what is the advantage of your solution?

-The implemented system is wearable but some important parameters as weight, dissipated power were not specified.

Some minor typos :

     In Figure 1  Rec should be changed Res;

     row 139 some missing reference was given.

Author Response

(The authors gave the same response as above.)

Round 2

Reviewer 1 Report

The author has responded to me with a modified experiment setup and a clear explanation.

Unfortunately, this work is more like a design report, without any design innovation or trade-off consideration. The contribution of this work, as I mentioned in the last review, is still not clear. 

I will highly suggest that the authors should give either a literature search plus discussion (on their advantages and disadvantages) or give design consideration/discussion for their systems, i.e. how to achieve high bandwidth driver for fast and accurate waveform output, what to consider on the power consumption reduction, etc. These will make this paper helpful for the readers.

Author Response

Thanks to the remarks of the reviewers, we revised manuscript entitled "Wearable High Voltage Compliant Current Stimulator For Sensory Feedback Restoring" and submit it for minor revisions.

Reviewer 2 Report

This work can be accepted.

Author Response

Thanks to the remarks of the reviewers, we revised manuscript entitled "Wearable High Voltage Compliant Current Stimulator For Sensory Feedback Restoring" and submit it for minor revision.

Reviewer 3 Report

The posed problem is interesting but the achieved results have to be compared to

(1) the real biological feed-back signals in literature and/or

(2) to validate the results on patients by measurements.

The contribution reported  a four-channel feed-back. The relationship between the number of channels and the accuracy of the feed-back have to be analysed.

Author Response

(The authors gave the same response as above.)
